# Altered Muscle–Brain Connectivity During Left and Right Biceps Brachii Isometric Contraction Following Sleep Deprivation: Insights from PLV and PDC

**DOI:** 10.3390/s25072162

**Published:** 2025-03-28

**Authors:** Puyan Chi, Yun Bai, Weiping Du, Xin Wei, Bin Liu, Shanguang Zhao, Hongke Jiang, Aiping Chi, Mingrui Shao

**Affiliations:** 1Department of Physical Education, Shanghai Maritime University, Shanghai 201306, China; chip2@coventry.ac.uk (P.C.); sgzhao@shmtu.edu.cn (S.Z.); hkjiang@shmtu.edu.cn (H.J.); 2Faculty of Business and Law, Coventry University, West Midlands CV1 5FB, UK; 3New Business School, Shaanxi Vocational and Technical College, Xi’an 710104, China; xiaotty@snnu.edu.cn; 4Sports Institute, Ningxia Normal University, Guyuan 756001, China; 82015006@nxnu.edu.cn; 5School of Software, Xi’an Jiaotong University, Xi’an 710049, China; starwei@stu.xjtu.edu.cn; 6School of Sports, Shaanxi Normal University, Xi’an 710119, China; 2006111@snnu.edu.cn; 7School of Sports, Shanghai Normal University, Shanghai 201418, China

**Keywords:** sleep, brain connectivity, isometric contraction, PLV, PDC

## Abstract

Insufficient sleep causes muscle fatigue, impacting performance. The mechanism of brain–muscle signaling remains uncertain. In this study, we examined the impact of sleep deprivation on muscle endurance during isometric contractions and explored the changes in brain–muscle connectivity. Methods: The research involved 35 right-handed male participants who took part in an exercise test that included isometric contractions of the left and right biceps in both sleep-deprived and well-rested states. Muscle contraction duration and electroencephalogram (EEG) and electromyography (EMG) signals were recorded. Functional connectivity between brain regions was assessed using the phase locking value (PLV), while partial directed coherence (PDC) was used to analyze signal directionality between motor centers and muscles. Results: The connectivity strength between Brodmann areas (BAs) 1-5 and the right BA6, 8 regions was significantly decreased in the isometric contractions after sleep deprivation. Insufficient sleep enhanced the PDC signals from the motor center of the right brain to the left biceps, and it decreased the PDC signals from both biceps to their opposite motor centers. Conclusions: Sleep deprivation shortened muscle isometric contraction duration by affecting the interaction between the somatosensory motor cortex and the right premotor cortex, reducing biceps feedback signal connectivity to the contralateral motor center in the brain.

## 1. Introduction

Brain functional fatigue refers to a temporary decline in brain function caused by long-term or high-intensity cognitive activities. Its mechanism is the excessive consumption and insufficient recovery of brain resources, and it serves as a protective signal indicating that rest and adequate sleep are required to restore the steady state of the nervous system [1]. During deep sleep, the body’s cells and tissues can effectively repair, grow, and function properly. Sleep also plays a crucial role in promoting brain functions such as protein synthesis and memory, as well as in removing harmful substances from the brain to protect it and maintain optimal function [2,3]. However, a lack of sleep or sleep disorders can lead to memory problems, difficulties in concentrating, and impaired cognitive abilities, as well as contributing to various physical illnesses [4]. In terms of causes, exercise fatigue is more complex than fatigue caused by a lack of sleep [5]. Endurance exercise leads to the depletion of energy substances [6], while anaerobic exercise results in the accumulation of lactic acid and the disruption of the internal environment [7].

An electroencephalogram (EEG) can be used to effectively detect the degree of central fatigue caused by endurance exercise. Fatigue induced by strength exercise usually manifests as peripheral fatigue, e.g., muscular fatigue and neuromuscular coupling fatigue [8,9]. The electromyogram (EMG) signal records the electrical activity produced during muscle contraction, representing the collective action potentials of motor units. Its amplitude and frequency are directly linked to neuromuscular functional activities [10]. Many studies have shown an increase in the time-domain indices and a decrease in the frequency-domain indices of EMG during fatigue caused by the isometric contraction of muscles [11,12]. Furthermore, studies have also confirmed an increase in the time-domain metrics and a decrease in the frequency-domain metrics of EMG after sleep deprivation [13]. The mean power frequency (MPF) is related to the synchronization of the type of motor units recruited and the discharge frequency of skeletal muscle during contraction, and the median frequency (MF) is mainly affected by the composition of the different myofiber types of muscle tissues; thus, MPF and MF are appropriate metrics for evaluating fatigue during muscle contraction.

After central fatigue, the EEG signal is mainly characterized by an increase in slow-wave components, such as δ and θ waves [14]. Similarly, exercise-induced fatigue occurs with corresponding changes in the EEG, especially an increase in the slow-wave component [15]. The above study suggests that the EEG signal is an important evaluation index for monitoring and reflecting fatigue in the brain in resting and working states. When the human body completes a motor task, aside from information exchange between brain regions, the transmission of information and feedback of motion-related brain regions and muscles are also important factors. Skeletal muscle contraction type, intensity (resistance load size), and duration all affect the occurrence of skeletal muscle fatigue, and EMG signals are an effective indicator of skeletal muscle contraction, reflecting the working state of skeletal muscle and allowing for the judgment of fatigue in real time [16]. EMG and EEG represent the electrical activities of muscle and the brain, respectively, and they are often used in the fields of motor control, neuromuscular diseases, sleep disorders, and brain–computer interfaces, reflecting the functional transmission and regulation process of the nervous system from the brain to the muscle [17].

A lack of sleep has a significant negative impact on the strength performance of athletes. Studies have shown that sleep deprivation significantly reduces the maximum strength output and explosive power of athletes [18]. Partial sleep deprivation (4 h/day) results in a significant reduction in athletes’ maximum strength and endurance [19], and sleep deprivation reduces neuromuscular control and increases the risk of sports injury [20]. In contrast, 7–9 h/adequate sleep (days) is a key factor in optimizing athletic performance and strength output [21]. In conclusion, sleep is a key factor in determining athletic performance, and optimizing sleep is very important for athletes to improve their strength performance. In athletes, sleep deprivation causes the accumulation of fatigue in the central brain, which, in turn, leads to a decrease in their exercise capacity and the early onset of exercise fatigue [22]. By analogy, sleep deprivation inevitably causes a decrease in the ability of athletes to perform isometric contractions; however, few studies have reported what types of change characteristics occur in brain EEGs and muscle EMGs in this process. Therefore, in the present study, we established a sleep deprivation model and an isometric contraction model of the biceps brachii muscle in athletes, and we monitored the changes in EMG and EEG signals. We analyzed the signal changes in the brain–muscle axis in the superposition of sleep-induced central fatigue and muscle contraction-induced motor fatigue, and we further elucidated the mechanism of the effects of sleep deprivation on exercise-induced fatigue.

## 2. Materials and Methods

### 2.1. Participants

A total of 40 college students from the physical education program at Shaanxi Normal University were recruited to participate in this experiment. They met specific criteria, including being healthy, having good-quality sleep, engaging in strength training 1–2 times a week, and having a healthy body fat percentage and body mass index (BMI). The exclusion criteria included cardiovascular disease, diabetes, and other diseases not suitable for exercise; smoking, drinking, coffee intake, vigorous exercise, and mood swings within 24 h before the experiment; accumulation of physical or mental fatigue before the experiment; insufficient sleep and poor sleep quality before the experiment; and serious mental illness or cognitive impairment. After screening, 35 male students were selected to participate. All participants provided informed consent, and the study was approved by the Ethics Committee of Shaanxi Normal University (No. 202416053). Basic information about the participants can be found in Table 1. An assessment of sharp hand judgement was conducted using the Edinburgh Handedness Inventory [23]. Body fat percentage and BMI were measured using a Body Composition Meter (Inbody 3.0, Seoul, Republic of Korea), while the 1-repetition maximum (RM) test for biceps was conducted following Huanhuan Guo’s method [24].

### 2.2. Sleep Deprivation and Exercise Modeling

The experiment on modeling sleep deprivation was based on the American Sleep Foundation’s recommendations for young people aged 18–25 years [25]. Severe sleep deprivation was defined as less than 4 h of sleep, while normal sleep was considered to be 7 h or more. In this study, sleep deprivation caused by staying up late was simulated with subjects who followed a normal daily routine without accumulating fatigue.

All subjects, as the same group of subjects, participated in two experimental models of sleep deprivation and normal sleep. In the sleep deprivation model, participants were instructed to remain awake until 3 am and were then roused at 7 am the following day, resulting in less than 4 h of sleep. Subsequently, they underwent testing using the Rating of Perceived Exertion (RPE) scale, with successful sleep deprivation characterized by a RPE score of 15 or above. Contrastingly, in the normal sleep model, participants were directed to retire to bed at 10 pm and awaken at 7 am the subsequent day, ensuring a sleep duration exceeding 7 h. A successful normal sleep model was identified by a RPE score below 10. The establishment sequence of the two sleep models follows a randomization principle, with a minimum interval of 24 h between the models to prevent mutual interference. The main parameters following the establishment of the sleep deprivation model can be found in the Appendix A (shown in Appendix A).

Upon the successful establishment of either sleep deprivation or a normal sleep pattern, participants engaged in a continuous isometric contraction test of either the left or right biceps brachii, with the order of biceps testing randomized. A 5–10 min break was provided between tests for electrode reapplication and participant rest. Each participant sat on a stool against a wall, with their upper body upright, their arms naturally hanging down, and their left or right arm palm facing up. They were then loaded at the wrist with a weight using an elastic band equivalent to 30% of their one-repetition maximum for biceps muscle contractions. The placement of the electromyography electrode was based on Surface ElectroMyoGraphy for the Non-Invasive Assessment of Muscles (SENIAM) project recommendations [26]. The hair on the skin where the electrodes were placed was shaved, and then cleaned with alcohol to ensure prolonged adhesion and reduced resistance [27]. The participants contracted their biceps muscle so that the elbow joint was at a 90-degree angle, and they maintained this position until they experienced fatigue exhaustion and stopped contracting. The order in which they performed the isometric contractions with their left and right biceps was randomized, with a minimum interval of 30 min between the two exercises. The criteria for determining exhaustion during the biceps isometric contraction included feeling soreness or trembling in the biceps, the inability to maintain a 90-degree angle at the elbow joint, and an RPE score of 19 or higher (indicating extreme tiredness). Further details on the sleep deprivation and exercise models can be found in Figure 1.

### 2.3. Acquiring and Processing EEG and EMG Data

Before performing the isometric contractions, the participants were fitted with a 68-channel electroencephalogram–myoelectric acquisition system from TMSi in the Netherlands. This system captured their EEG signals during both resting periods and isometric contractions. The resistance between the electrodes and the scalp was reduced to less than 10 kΩ. A bipolar channel of the device, connected to electrode pads on the subject’s left or right arm biceps muscle belly, recorded their EMG signals at a sampling frequency of 1000 Hz. The electrodes were placed according to the international 10-10 system [28]. Additionally, the duration of the contractions of the biceps in the left and right arms was recorded.

The processing flow of the EEG and EMG data is outlined in Figure 2. The EEGLAB toolkit (Version R2022b, San Diego, CA, USA) was loaded into MATLAB software (Version R2022b, MathWorks, Natick, MA, USA). The data underwent batch processing with the following main steps: channel localization, extraction of useful electrodes, segmentation into epochs (2 s per epoch), removal of bad data epochs, and separation of EEG and EMG data from exercise data. The EEG data then underwent band-pass filtering (1 to 100 Hz), with two notch filters (48 to 52 Hz and 98 to 102 Hz), to eliminate the interference of alternating current frequencies, along with a sampling rate adjustment to 500 Hz, and re-referencing to the bilateral mastoid process (M1 and M2); additionally, independent component analysis (ICA) was applied in EEGLAB, and ICLable was loaded to remove artefactual components such as blinks, eye movements, head movements, and cardiac activity [29]. Finally, clean EEG data were obtained for further analysis. Conversely, EMG data were consolidated into continuous data, converted to mat file format, and subjected to two band-pass filters (20 to 150 Hz) for an analysis of the root mean square (RMS), MPF, and MF.

### 2.4. Analysis of Functional Connectivity of ROIs

The brain regions of interest (ROIs) selected for this study were those related to the sensorimotor center, including the premotor cortex, motor auxiliary cortex, and primary motor sensory cortex [30]. The 10-10 electrode position closest to each Brodmann area (BA) was identified [31], and the names of the BAs and their corresponding channel names are listed in Appendix A. Brainstorm software (version 3.231220, USC & McGill University) was installed on MATLAB, as described in a previous study [32]. The processed clean EEG data were then imported into Brainstorm, where the phase locking value (PLV) was chosen as the measure of functional connectivity. The PLV indicates the phase difference between the discharge signals in two brain regions, X and Y, thus providing information on the phase coupling between the two EEG signals. The data were preprocessed with band-pass filtering and Hilbert transform before analysis.

Brainstorm’s interface operations involve the following steps: first, matching the locations of the EEG channels to the ASA-10-5 standard channels; then, calculating noise covariance and data covariance; and, finally, establishing the head and source models. In the Brainstorm operation interface, a total of 13 brain regions, including both the left and right brain hemispheres, were selected to correspond to the specified ROIs (note that BA3(R) is lacking in Brainstorm’s scout template). Finally, the PLV was calculated between the selected ROIs. After using Brainstorm software to calculate the matrix data of the PLVs between the ROIs for each subject, the data were imported into GRETNA (version 2.0.0) for further analysis [33]. Intra-group differences were calculated, and the network-based statistic (NBS) method was used for multiple comparison correction [34]. Additionally, using the PLV matrix data, global network attributes of 13 ROIs were calculated, including small-world attribute sigma, global efficiency (E_g_), and local efficiency (E_loc_). The small-world attribute is used to measure whether a network has small-world characteristics. It is generally believed that sigma > 1 indicates that the calculated matrix has small-world characteristics [35]. *E_g_* is a measure of the overall efficiency of information transfer in a network and is defined as the reciprocal average of the shortest path lengths between all pairs of nodes. *E_loc_* is a measure of the efficiency of the neighbor subgraph of each node in the network and reflects the fault tolerance of the network, thus indicating the speed of information transmission in a complex network. To measure the transmission capacity of local information, the calculation formula is as follows:$$\sigma =\frac{\gamma }{\lambda }=\frac{C_{actual}/C_{random}}{L_{actual}/L_{random}}$$
where γ is the normalized clustering coefficient of the network, *C_actual_* is the clustering coefficient of the actual network, and *C_random_* is the clustering coefficient of the random network. λ is the normalized average path length of the network and *L_actual_* is the average path length of the actual network. *L_random_* is the average path length of a random network.$$E_{global}=\frac{1}{N(N-1)}\sum_{i\neq j}\frac{1}{d_{ij}}$$
where *N* is the number of nodes in the network and $d_{ij}$ represents the shortest path length between nodes *i* and *j*.$$E_{loc}=\frac{1}{n}\sum_{i}E_{loc}(i)=\frac{1}{k_{i}{(k}_{i}-1)}\sum_{j,h\in N_{i}}\frac{1}{d_{jh}}$$
where Eloc(i) is the local efficiency of node *i*, n indicates the total number of nodes, $k_{i}$ indicates the degree of node *i*, $N_{i}$ is a set of neighbor nodes of node *i*, and $d_{jh}$ is the shortest path length between nodes *j* and *h*.

The parameters set during the analysis of the network attributes were as follows: a relative threshold as the thresholding method; a sparsity range of 0.22–0.29, with a step size of 0.01; 1000 random networks; and a binary clustering coefficient type. BrainNetViewer software (version 1.7) was utilized to visualize the functional connectivity differences in the brain regions. A connection drawing in space was generated using the SurfTemplatefile BrainMesh_ICBM152 [36].

### 2.5. Analysis of Effective Connectivity Between EEG and EMG

Brodmann areas 1 to 4 are functional brain regions responsible for primary motor and sensory functions in the human body. These areas play a crucial role in the efferent transmission of neural impulses and the afferent transmission of sensory signals during muscle contraction [37]. Among the channels corresponding to the Brodmann areas, C3 and C4 are key channels in the primary motor sensory area of the brain; they represent and reflect EEG signals from the left and right primary motor sensory areas, respectively. Therefore, C3 and C4 were selected as the channels of interest (COIs) from the 30 channels in the clean EEG data in order to determine the lateralization characteristics of the primary motor sensory center. Additionally, the EMG data were integrated into the EEG data containing only C3 and C4. After combining the data, 180 s segments were uniformly extracted and processed continuously in preparation for a partial directed coherence (PDC) analysis. The HERMES toolbox (version 2020-04-26, https://hermes.med.ucm.es/) was loaded into MATLAB for the analysis, with PDC chosen as the measure of effective connectivity [38]. PDC, a frequency-domain metric rooted in Granger causality, is derived from multivariate autoregressive processes modeling time series. The calculation of PDC focused on the EEG signals from channels C3 and C4 in relation to the EMG signals from either the left or right biceps muscle. The calculation formula of PDC is as follows:$${PDC}_{ij}\left(f\right)=\frac{{\left|H_{ij}(f)\right|}^{2}}{\sqrt{\sum_{k=1}^{N}{\left|H_{kj}(f)\right|}^{2}}}$$
where ${PDC}_{ij}\left(f\right)$ represents the intensity of information flow from j to i, and $H_{ij}(f)$ is an element of the transfer function matrix. PDC ranges between 0 and 1, with values closer to 1 signifying increased information transfer.

### 2.6. Data Statistics

SPSS (Version 27.0; SPSS, Inc., Chicago, IL, USA) was utilized to analyze the duration of isometric contractions, and RMS, MPF, and MF values. Z-scores were computed to assess outliers, with values exceeding ±3 being considered extreme. Subsequently, extreme values were replaced using the method of sequential means. Normal distributions were assessed using the Shapiro–Wilk method. Data following a normal distribution are presented as mean ± standard deviation. The isometric contraction duration and slow-to-fast wave ratio were compared using a paired samples *t*-test. Global property indicators such as the sigma, E_g_, and E_loc_ values in different sport states were compared using a repeated-measures ANOVA in a two-way analysis (sleep deprivation vs. different sport states). Non-normally distributed data are presented as median values (upper and lower quartiles) and were compared using Wilcoxon’s test for two groups and Friedman’s test for three groups. The PDC values from the HERMES toolkit were averaged, and false discovery rate (FDR) correction was applied for multiple comparisons. A significance level of *p* < 0.05 was used for the statistical analysis. The PLV values of brain regions under different frequency bands were compared using network-based statistic (NBS) correction in the GRETNA toolkit, and the *p*-value thresholds for both edges and components were set at 0.05, with 2000 permutation tests conducted.

## 3. Results

### 3.1. Results of Biceps Isometric Contraction Time and EMG Index

A total of 35 participants performed isometric contractions of the biceps brachii muscle in both arms following a period of sleep deprivation, and the collected data passed a test for normal distribution; thus, the comparison was conducted using a *t*-test. The results, as displayed in Figure 3, demonstrated a noticeable difference in time to exhaustion between the two conditions. It was observed that the duration of muscle contractions in the left and right arms was considerably shorter after sleep deprivation than after normal sleep (*p* < 0.001). These findings suggest that inadequate sleep can have a negative impact on muscle performance.

As shown in Figure 4A–F, parametric tests were used for comparison due to the data not meeting the normal distribution assumption. The results indicate that sleep deprivation did not significantly affect the RMS or MPF values in different bands of both the left and right biceps. However, there was a notable decrease in the MF value of the left biceps in the high-frequency band of the EMG (100 to 150 Hz) after sleep deprivation (*p* < 0.05), while the MF value of the right biceps remained unchanged. Additionally, the RMS of the left biceps was significantly higher than that of the right biceps, and the MPF was lower than that of the right biceps (*p* < 0.05) in the low-frequency band of 20–100 Hz. These findings indicate that both central and peripheral fatigue resulting from sleep deprivation have a notable impact on the MF in the high-frequency band during isometric muscle contraction. However, there appears to be a discrepancy in the low-frequency band based on handedness.

### 3.2. Comparison Results of Global Properties Among ROIs

The premise of a network attribute analysis is that the sigma value is greater than 1. According to this condition, the network properties of the PLV matrix numbers in the δ, α, and β bands were analyzed in this study, and the analysis results are shown in Appendix A. In contrast to adequate sleep, the effects of sleep deprivation on the global attribute values of the ROIs were concentrated in the α frequency band. The difference in the alpha band is shown in Figure 5.

Figure 5 shows a non-significant sigma interaction effect between the two sleep qualities and the three physical motion states (F (2, 68) = 1.570, *p* = 0.215, ɳ^2^ = 0.044). However, the main effect of the physical motion states was found to be significant (F (2, 68) = 5.473, *p* = 0.006, ɳ^2^ = 0.139). Further analysis through a paired comparison indicated that the sigma value of the right biceps isometric contraction was significantly higher than that of the resting state before contraction (*p* = 0.001).

Next, it was found that the interaction effect of E_g_ between the two sleep qualities and the three physical activity states was significant (F (2, 68) = 5.146, *p* = 0.008, ɳ^2^ = 0.131). The simple effects of the different sleep conditions (F (2, 33) = 17.716, *p* < 0.001, ɳ^2^ = 0.518) and the different states (F (1, 34) = 13.949, *p* = 0.001, ɳ^2^ = 0.291) were significant. After good sleep, the E_g_ values were significantly lower in both the left and right biceps during contraction (*p* = 0.024 and *p* < 0.001) than during the resting state (*p* = 0.024 and *p* < 0.001), but they were significantly higher in the right biceps after sleep deprivation than after good sleep (*p* = 0.001).

In addition, the interaction effect of E_loc_ between the two sleep qualities and the three physical activity states was significant (F (2, 68) = 3.495, *p* = 0.036, ɳ^2^ = 0.093). The simple effects of the different sleep conditions (F (2, 33) = 10.567, *p* < 0.001, ɳ^2^ = 0.390) and the different states (F (1, 34) = 5.996, *p* = 0.020, ɳ^2^ = 0.150) were significant. After good sleep, the E_loc_ value of the right biceps during contraction was significantly lower than that during the resting state (*p* < 0.001), and, compared with good sleep, the E_loc_ value of the sleep-deprived subjects was significantly higher in the resting state (*p* = 0.020).

### 3.3. Comparison Results of Functional Connectivity Among ROIs

This experiment included the selection of ROIs, as shown in Figure 6A. After a restful sleep, the participants performed isometric contractions of their left and right biceps. The changes in the PLVs between the ROIs are depicted in Figure 6C,E. During both the left and right biceps contractions, there was an increase in connectivity between BA8 (R) and the left primary sensorimotor area (*p* < 0.05). Notably, when the left biceps was contracted, BA8 (R) also showed increased connectivity with BA1, 2, 4, and 5 (R) on the same side (*p* < 0.05). Similarly, during the right biceps contraction, BA8 (R) exhibited increased connectivity with BA1, 2, 4, and 5 on the left side (*p* < 0.05). Additionally, BA5 (L) displayed strong connectivity with BA1, 2, and 4 on the same side. Following sleep deprivation, there was a significant decrease in the PLV between the left and right hemispheres during the resting state. This decrease was mainly observed in the correlation between BA1, 2, 3, and 4 in the left hemisphere and BA1, 2, 4, and 5 in the right hemisphere (*p* < 0.05), as illustrated in Figure 6B.

Furthermore, as depicted in Figure 6D,F, the participants performed isometric contractions of their left and right biceps following a period of sleep deprivation. The PLVs of the brain regions mentioned exhibited significant variations after the isometric contractions. Specifically, during the contraction of the left biceps, there was a notable enhancement in connectivity between the primary motor sensory areas of the left and right hemispheres. This was evidenced by the increased connectivity between BA1, 2, 4, and 5 on the left with BA1, 2, 4, and 6 on the right (*p* < 0.05). In contrast, the connectivity of BA1, 2, 3, and 4 on the left did not show significant enhancement during the contraction of the right biceps (*p* < 0.05). These findings suggest that sleep deprivation has differential effects on brain region connectivity during isometric contractions based on handedness.

### 3.4. Results of EEG-EMG During Isometric Contractions

The statistical analysis in Appendix A revealed that, in the common frequency range of the EMG and EEG (13–100 Hz), the relationship between the EMG signals of the left and right biceps and the signals of the primary motor sensory cortex in the brain significantly changed after sleep deprivation compared to after good sleep. Changes in the EEG-EMG signals during the isometric contraction of the left biceps were mainly observed in the γ^1^ range (30–60 Hz), as depicted in Figure 7A. There was a notable increase in muscle PDC signals to the left cerebral region (*p* = 0.037) and a decrease in PDC signals to the right cortex (*p* = 0.013). The PDC signals from the left cortex to the muscle were significantly weakened (*p* = 0.012), while the PDC signals from the opposite cortex were enhanced (*p* = 0.014). Additionally, the PDC data from the left cortex to the right brain region decreased, while the signals from the right brain region to the left cortex increased. For isokinetic contractions of the right biceps, post-sleep deprivation, the PDC of the muscles sending signals to the left and right primary motor sensory areas significantly decreased only within γ^2^ (60–100 Hz) (*p* = 0.033 and 0.021), and the PDC from the right brain area to the contralateral brain area also significantly decreased (*p* = 0.019), as shown in Figure 7B.

## 4. Discussion

It is well documented that sleep deprivation contributes to central fatigue in athletes, leading to a reduced exercise capacity and early fatigue onset [39]. Sleep deprivation not only decreases an athlete’s maximum power output but also weakens their strength endurance performance [40]. This study also confirmed that sleep deprivation results in a significant decrease in the duration of force during isometric contractions of the biceps brachii muscle, indicating that insufficient sleep negatively impacts muscle fatigue in these contractions. Research has suggested that sleep deprivation may decrease the excitability of cortical and spinal motor neurons [41]. Sleep deprivation can lead to a decrease in the synthesis and release of neurotransmitters such as dopamine and serotonin, which, in turn, impacts the recruitment efficiency of motor units and the coordination of muscle contractions [42]. Inadequate sleep can also lead to an imbalance in the autonomic nervous system, resulting in decreased muscle endurance [43]. From an energy metabolism perspective, sleep deprivation markedly reduces the muscle glycogen storage capacity, consequently impacting strength and endurance performance [44]. Therefore, the mechanisms underlying the negative impact of sleep deprivation on muscle endurance are multifaceted.

The effects of sleep deprivation on the EMG of the biceps brachii muscle during isometric contractions were investigated in this study. The results show no significant effect on the RMS. Sleep deprivation typically results in a decreased recruitment capacity of motor units, leading to a significant decrease in RMS values, particularly evident during high-intensity exercise [45]. However, some studies have shown that there is no significant change in the RMS during isometric contractions of equal length, which may be attributed to the relatively stable recruitment pattern of motor units and the consistent discharge characteristics of slow-twitch muscle fibers [46]. Especially during low- or moderate-intensity isometric contractions, changes in the RMS may not be immediately apparent [47]. Some studies have shown a significant impact of sleep deprivation on the frequency-domain indicators MPF and MF during isometric contractions of skeletal muscles. Research has found that sleep deprivation markedly reduces the values of MPF and MF, especially during prolonged isometric contractions [48]. This may be related to the increased sensitivity of frequency-domain indicators to changes in muscle fatigue and neural control [49]. This study confirms a reduction in the MF in the left biceps from the 100 to 150 Hz range following sleep deprivation. The results indicate that sleep deprivation affects the MF in the high-frequency band during muscle contractions. Our above experimental results are similar to those of Zhou et al. [49]. The mechanism may involve sleep deprivation reducing the excitability of the cerebral cortex and spinal motor neurons, decreasing the high-frequency discharge of motor units, thereby resulting in a significant decrease in MF values in the higher-frequency range [50]. To sum up, this change suggests that decreased central nervous system excitability and accelerated muscle fatigue influence the discharge frequency of motor units.

In this study, the global attribute values of ROIs were compared based on PLV matrix data. Significant differences were found in the α frequency band for sleep quality and body motion states but not in other frequency bands. Therefore, in this study, the sigma value in the alpha band was chosen to represent the small-world property, global efficiency, and local efficiency of the ROIs. The results suggest that isometric muscle contraction exercises significantly influence muscle–brain communication. During motor tasks, the overall collaborative efficiency and local information processing efficiency of brain regions were enhanced. This suggests that coordination between different brain regions is crucial for isometric muscle contractions. However, global efficiency decreases during isometric muscle contractions, possibly due to the specialization and differentiation of tasks among different brain regions, leading to a decrease in overall efficiency. This indicates that the brain functions differently during various motor tasks. Furthermore, a lack of sleep also impacts the brain’s functioning. Research has indicated that individuals with infantile spasms may experience improvements in overall efficiency as brain functional networks compensate for decreased local efficiency [51].

Sleep deprivation has been shown to weaken the small-world properties of brain networks, demonstrating an imbalance in local and global functional integration [52]. Moreover, sleep deprivation leads to a substantial reduction in the global efficiency of brain networks [53]. During isometric contraction, there is an acceleration in the transmission speed of information between different regions of the brain, leading to an increase in the global efficiency of brain networks [54]. During isometric contractions, changes in brain network properties reflect dynamic adjustments in muscle control. In this study, the participants who performed isometric contractions after sleep deprivation showed no significant changes in small-world attributes, global efficiency, or local efficiency, which is a result of the combined effects of these two factors. Following sleep deprivation, the central nervous system maintains the stability of the brain network through compensatory mechanisms, especially during low-intensity exercise [18]. The slight alterations in brain network characteristics during low-intensity physical activity may be linked to the compensatory capabilities of the central nervous system [55]. This study’s findings revealed that there was a compensatory rise in overall efficiency during isometric muscle contractions following sleep deprivation, indicating that the brain can still effectively carry out motor activities despite being sleep-deprived.

PLV analysis is a valuable tool for studying how different brain regions synchronize and communicate with each other [56]. This technique helps researchers understand how the brain coordinates during different states. After sleep deprivation, functional connectivity between several brain areas related to movement, such as BA1-5, across the left and right hemispheres significantly decreases. This could be due to insufficient sleep weakening the strength of connections between neurons, as well as sleep deprivation leading to reduced levels of neurotransmitters such as dopamine and glutamate, thereby affecting the efficiency of neural signal transmission [57]. The findings of this study show that, while performing contractions with both the left and right biceps, there was an increase in connectivity between BA8 (R) and the primary sensorimotor areas on both sides. BA8 is primarily associated with conscious motor control, with the right BA8 region specifically linked to memory retrieval. Our experimental results demonstrate an interesting observation: there was a notable increase in functional connectivity between the sensorimotor brain regions (e.g., BA1, 2, 3, 4, and 5) and the right BA8 region during isometric contractions of the biceps brachii in both the left and right arms, while no connectivity was observed with the left BA8 region. In addition to engaging in advanced cognitive tasks such as planning, decision-making, and problem-solving, BA8 is also involved in the planning and coordination of complex movements, particularly in tasks requiring fine motor control [58]. Both BA8 regions exhibited overlap and differences in function. The left BA8 region is involved in language production, planning, and the execution of complex cognitive tasks, while the right BA8 region plays a crucial role in visual motor coordination, eye movement control, and complex motor tasks [59]. In this experiment, the participants were required to perform a contraction action of the biceps brachii muscle, maintaining a fixed position of elbow flexion at 90 degrees. The right BA8 region plays a role throughout the entire movement, thus showing enhanced connectivity with sensorimotor centers.

Therefore, in the participants in this experiment, the connection between BA8 and the sensorimotor brain areas was engaged in order to sustain the posture of biceps flexion. The results of this experiment show that sleep deprivation significantly reduced the functional connections between motion-related brain areas, particularly the left and right primary sensorimotor centers, under quiet conditions. This decline in brain area connections due to a lack of sleep is supported by many studies [60,61]. Insufficient sleep not only weakens the information processing efficiency of the sensorimotor network but also reduces the collaborative capacity between the right BA8 region and the sensorimotor center. In short, sleep deprivation significantly impairs the functional connectivity between the sensorimotor center and right BA8 region during movement. It is important to recognize that handedness can have varying effects on neurotransmission and brain function [62]. Despite the similarities in the hand motor network between left- and right-handed people, there are noticeable differences in the asymmetry of their connectivity patterns [63]. Unfortunately, we did not recruit any left-handed participants for this experiment, resulting in a lack of evidence from left-handed individuals. For right-handed participants, their dominant and non-dominant arms were more appropriately described as strong- and weak-side arms. Our findings revealed differences in brain functional connectivity between the left and right arms; specifically, compared to the biceps brachii contraction in the right arm, the connectivity between the BA1-5 region on the right side of the brain and BA6 and BA8 was significantly enhanced during isometric contraction in the left biceps brachii under conditions of adequate sleep. Similarly, during isometric contraction under sleep deprivation, the interhemispheric connectivity of BA1-5 was significantly increased. These results support the notion that the non-dominant arm (weak-side arm) may exhibit neurocompensation. Numerous studies have also demonstrated neurocompensation in non-dominant arms during motor tasks [64].

PDC uses a multiple autoregressive model to assess how one brain region influences another, considering all other regions. It operates in the frequency domain, identifying causal relationships across frequencies and providing insights into information flow and functional integration in brain networks. In this study, the PDC outcomes from the signals of C3 and C4, which correspond to the left and right motor brain regions, were compared with the signals from the biceps muscle to analyze alterations in information transmission between the brain and muscles. It is well known that the sensory–motor centers control and provide feedback to the muscles of the limbs in a cross-over manner [65]. This study focused on the signal transmission relationship between the biceps brachii muscle and the contralateral sensorimotor cortex. During isometric contractions following sleep deprivation, there was no significant change in the PDC of information transmission between the left and right biceps brachii muscles and the C3 and C4 regions in the β frequency band (13–30 Hz). However, in the γ range (30–100 Hz), notable changes were observed. Specifically, during isometric contractions following sleep deprivation, the signal transmission from the left biceps brachii muscle to the contralateral sensorimotor brain region significantly decreased in the γ1 range (30–60 Hz), while from the right biceps brachii muscle, it decreased in the γ2 range (60–100 Hz). This indicates a negative impact of sleep deprivation on the feedback of muscle-to-central nervous system information during isometric contractions. Furthermore, an analysis of the sleep deprivation effects on the downward signal transmission from the brain to the muscles revealed a significant increase in the PDC signal from the C4 region representing the right sensorimotor cortex to the left biceps brachii muscle in the γ1 range (30–60 Hz), while the PDC signal from C3 to the right biceps brachii muscle showed no significant change. This suggests a compensatory mechanism of the sensorimotor cortex for the weak-side muscles.

The changes observed in the brain–muscle communication patterns indicate that a lack of sleep has a significant impact on the way that muscles and the motor cortex interact [66]. This altered communication may be due to the different strategies used by the body to compensate for the effects of sleep deprivation. For example, the increased communication from the muscles to the cortex in the left arm could be a way for the body to maintain motor function despite decreased control from the brain [67]. Additionally, the heightened communication from the right side of the brain to the left side suggests an effort to utilize extra neural resources to sustain motor performance. The specific frequency changes observed in different muscle groups imply that sleep deprivation affects various neural pathways involved in controlling movement [68].

## 5. Conclusions

This study found that inadequate sleep reduces the functional connectivity between the left and right sides of movement-related brain areas such as BA1-5. Furthermore, sleep deprivation weakens the functional connectivity between BA1-5 and the right BA8 region during the contraction of the biceps brachii muscle, leading to decreased information feedback from the movement-related brain areas to the contralateral side, ultimately resulting in a significant reduction in the duration of muscle isometric contraction. Therefore, inadequate sleep negatively impacts muscle strength and endurance due to the decreased functional connectivity in the movement-related brain areas caused by sleep deprivation, weakening the feedback of information from muscles to the sensory–motor brain centers. Additionally, after sleep deprivation, during weak arm biceps brachii isometric contractions, compensatory neural mechanisms were observed in the intracerebral functional connectivity of movement-related brain areas and the descending signal transmission from the brain to the muscles. However, this study lacks evidence regarding left-handed individuals, thus precluding an assessment of potential differences between left- and right-handed individuals based on the results obtained.

## Figures and Tables

**Figure 1 sensors-25-02162-f001:**
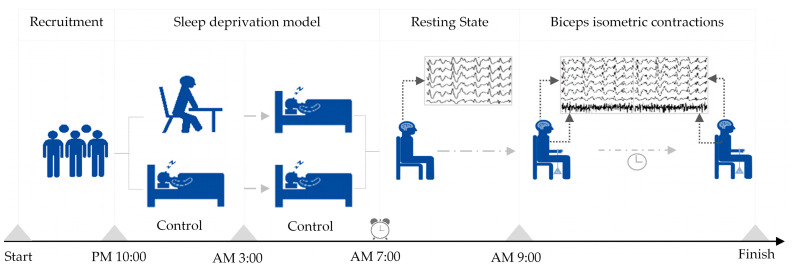
Establishment of sleep deprivation and exercise models.

**Figure 2 sensors-25-02162-f002:**
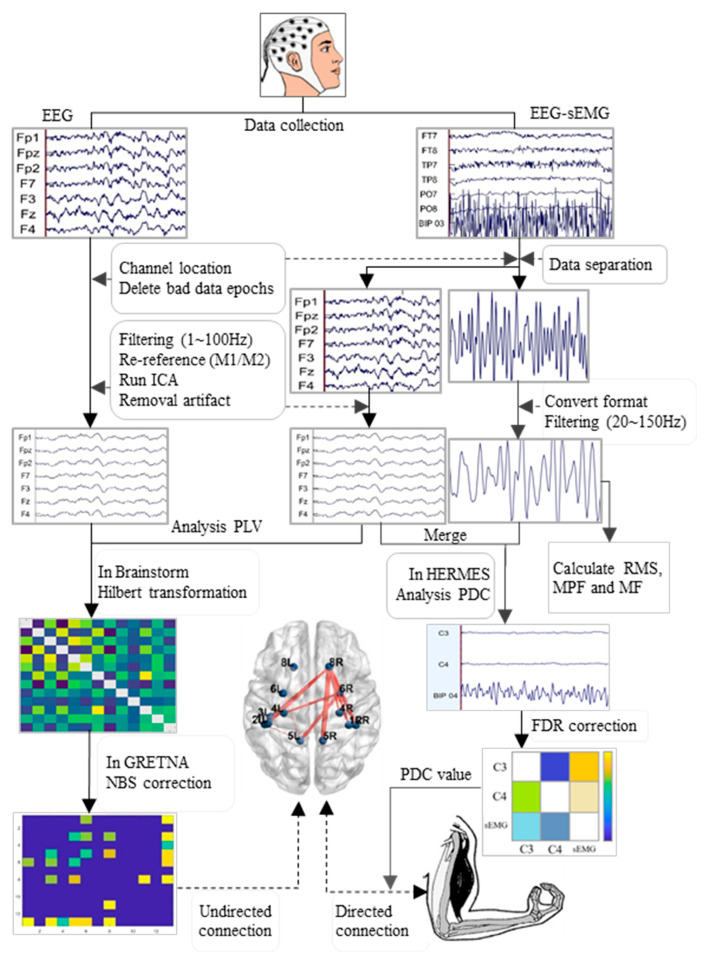
Flowchart of the processing and analysis of EEG and EMG data. EMG: electromyography; RMS: root mean square; MPF: mean power frequency; MF: median frequency; PLV: phase locking value; PDC: partial directed coherence; FDR: false discovery rate; NBS: network-based statistic.

**Figure 3 sensors-25-02162-f003:**
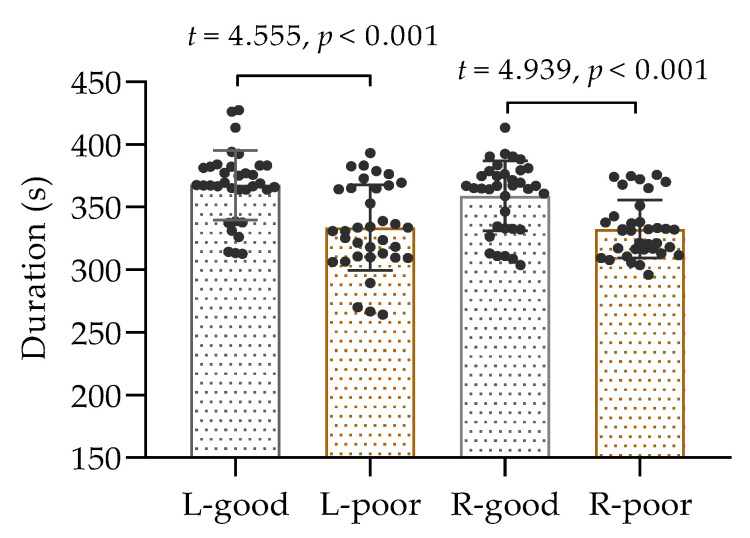
The duration of isometric contractions of the left and right biceps. L: left biceps, R: right biceps, good: after good sleep, poor: after poor sleep.

**Figure 4 sensors-25-02162-f004:**
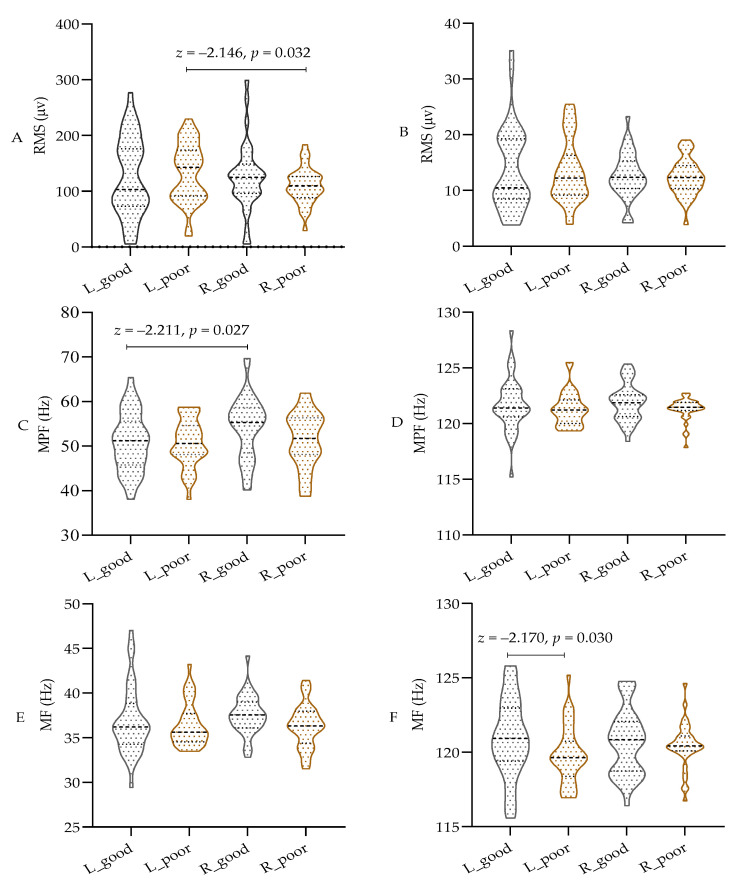
The test results of EMG in different frequency bands. (**A**) RMS in 20–100 Hz; (**B**) RMS in 100~150 Hz; (**C**) MPF in 20–100 Hz; (**D**) MPF in 100~150 Hz; (**E**) MF in 20–100 Hz; (**F**) MF in 100~150 Hz; RMS: root mean square; MPF: mean power frequency; MF: median frequency; L: left biceps, R: right biceps, good: after good sleep, poor: after poor sleep.

**Figure 5 sensors-25-02162-f005:**
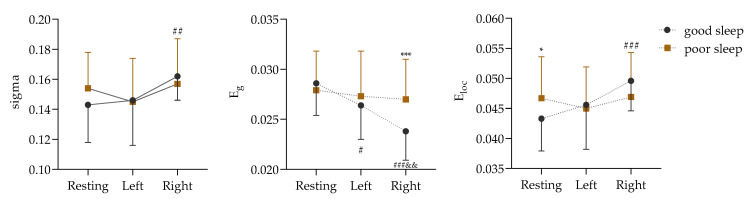
Comparison of global attributes in α frequency band based on PLV matrix data (Left: in isometric contractions of the left biceps. Right: in isometric contractions of the right biceps; vs. good sleep, *: *p* < 0.05, ***: *p* < 0.001; vs. resting, #: *p* < 0.05, ##: *p* < 0.01, ###: *p* < 0.001; vs. Left, &&: *p* < 0.01).

**Figure 6 sensors-25-02162-f006:**
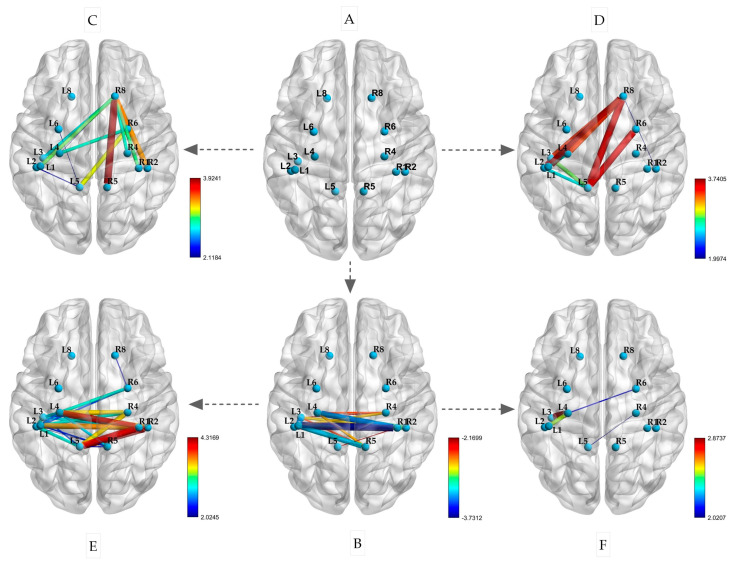
Changes in PLVs between ROIs during isometric contraction after sleep deprivation ((**A**) spatial locations of ROIs; (**B**) resting state after sleep deprivation; (**C**) left biceps contraction; (**D**) right biceps isometric contraction; (**E**) left biceps contraction after sleep deprivation; (**F**) right biceps isometric contraction after sleep deprivation; the size of the edge is based on the *t*-value).

**Figure 7 sensors-25-02162-f007:**
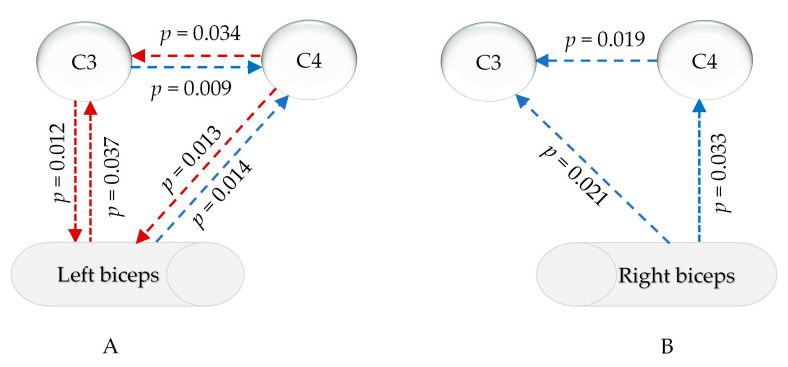
PDC changes in myocerebral signals after sleep deprivation (**A**) during left biceps contraction in the γ1 range (30–60 Hz); (**B**) during right biceps contraction in the γ2 range (60–90 Hz); red arrow: increased, blue arrow: decreased; *p*: *p*-value of FDR correction in comparing sleep deprivation with sleep sufficiency).

**Table 1 sensors-25-02162-t001:** Participants’ demographic characteristics.

Index	Results
Gender (number)	Male (35)
Age (years)	18~25
BMI (kg/m^2^) ^1^	22.02 ± 2.14
Percentage of body fat (%)	13.92 ± 4.63
1 RM value of the biceps of the left arm (kg) ^2^	14.95 ± 2.52
1 RM value of the biceps of the right arm (kg)	15.37 ± 3.29

^1^ BMI (body mass index) = weight/height2 (kg/m^2^). ^2^ RM: repetition maximum.

## Data Availability

Data from this study are available from the corresponding author.

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
