# Peer review of "Altered Muscle–Brain Connectivity During Left and Right Biceps Brachii Isometric Contraction Following Sleep Deprivation: Insights from PLV and PDC"

_sensors, 2025, doi:10.3390/s25072162_

Round 1
Reviewer 1 Report
Comments and Suggestions for Authors
The study focused on electrophysiological changes in the nervous system during muscle contraction in the context of sleep deprivation. By recording EMG and EEG during isometric contractions in different sleep states, the researchers aimed to reveal the potential impact of sleep deprivation on motor neural control mechanisms. The study analyzed the changes of functional connections between key brain regions such as motor cortex and sensorimotor cortex, and explored the mechanism of the impaired motor performance caused by sleep deprivation. Although the research perspective is novel, there are still many problems.
Minor problem:
1. MVC can standardize the value of EMG index. This standardization method can eliminate the difference of muscle strength and activation level between individuals and make the EMG data of different subjects more comparable. So as to improve the scientific and reliability of the research results. In the author's research method, the failure to adopt MVC standardization may lead to certain limitations in the interpretation and comparison of EMG data. Ask the author to provide reasons?
2. Line 115-117, the author mentions that the subjects tested 30% repetition maximum for biceps muscle contractions, and these data need to be supplemented.
A total of 14 brain regions were selected by the author in line 235. Why is one brain region missing in Line 230 and Figure 4?
4. Line 259-260, Global property indicators Key parameters for testing need to be described.
5. Line 266-267, Network-Based Statistic correction details need to be explained, such as what is the P-value threshold of significant edge? What is the P-value threshold for components?
6. PLV and PDC are the main test indicators in this paper. It is suggested that the author provide calculation formulas for PLV and PDC
7. In Figure 4, I think the author should provide a color scale for each function link diagram, which will be more standardized.
Major problem:
1 In the study of the neural mechanisms of motor control and postural regulation, the brain regions involved include: the prefrontal cortex, the sensorimotor cortex, the basal ganglia, the cerebellum, and the parietal lobe. In this study, only sensorimotor related brain regions were selected. Why not select whole brain regions to study from the perspective of brain network?
2. The frequency band related to fatigue is mainly theta frequency band, while alpha frequency band is selected for network attribute analysis in this study, why other frequency bands are not analyzed?
3. How does lack of sleep affect exercise? I feel that what the author said in the conclusion is not clear and thorough!
Author Response
Reviewer #1:
The study focused on electrophysiological changes in the nervous system during muscle contraction in the context of sleep deprivation. By recording EMG and EEG during isometric contractions in different sleep states, the researchers aimed to reveal the potential impact of sleep deprivation on motor neural control mechanisms. The study analyzed the changes of functional connections between key brain regions such as motor cortex and sensorimotor cortex, and explored the mechanism of the impaired motor performance caused by sleep deprivation. Although the research perspective is novel, there are still many problems.
Comment 1. MVC can standardize the value of EMG index. This standardization method can eliminate the difference of muscle strength and activation level between individuals and make the EMG data of different subjects more comparable. So as to improve the scientific and reliability of the research results. In the author's research method, the failure to adopt MVC standardization may lead to certain limitations in the interpretation and comparison of EMG data. Ask the author to provide reasons?
Response 1:Thank you for your comments. In this experiment, we selected 35 subjects, all of whom were college students majoring in sports, with little difference in physical conditions. We tested the MVC of their left and right biceps muscles before the experiment, and the data differences were not significant. We have upload MVC data to table 3 in Supplemental file 1.
Comment 2: Line 115-117, the author mentions that the subjects tested 30% repetition maximum for biceps muscle contractions, and these data need to be supplemented.
Response 2: Thanks for your suggestion, we add the data to table 2 in Supplemental file 1.
Comment 3: A total of 14 brain regions were selected by the author in line 235. Why is one brain region missing in Line 230 and Figure 4?
Response 3: Yes, in brainstorm, the scout template we selected is the Brodman brain template, which is missing the BA3 region on the left, so we can only get information about 13 brain regions.
Comment 4: Line 259-260, Global property indicators Key parameters for testing need to be described.
Response 4: Thank you for your comments. We have added the key parameters in the test of Global property indicators.
Please see line 253-278 in the revised manuscript.
Comment 5: Line 266-267, Network-Based Statistic correction details need to be explained, such as what is the P-value threshold of significant edge? What is the P-value threshold for components?
Response 5: Thank you for your comments and suggestions, We set the p-value thresholds for both edges and components were established at 0.05, The number of permutation tests was 2000 in Network-Based Statistic (NBS) correction. We added these parameter information. Please see line 320-321 in the revised manuscript.
Comment 6: PLV and PDC are the main test indicators in this paper. It is suggested that the author provide calculation formulas for PLV and PDC.
Response 6: Thank you for your comments. We have added the calculation formulas of several indexes in PLV and brain network attributes, please see line262-271 in the revised manuscript; In addition, we also added the calculation formula of PDC. Please see line 298-302 in the revised manuscript.
Comment 7: In Figure 4, I think the author should provide a color scale for each function link diagram, which will be more standardized.
Response 7: Thank you for your comments, we have added the color scale of the functional link chart. Please see Figure 6 in the revised manuscript.
Comment 8: In the study of the neural mechanisms of motor control and postural regulation, the brain regions involved include: the prefrontal cortex, the sensorimotor cortex, the basal ganglia, the cerebellum, and the parietal lobe. In this study, only sensorimotor related brain regions were selected. Why not select whole brain regions to study from the perspective of brain network?
Response 8: Thank you for your comments. Indeed, in the study of the neural mechanism of motor control and postural regulation, the brain regions involved include the prefrontal cortex, sensorimotor cortex, basal ganglia, cerebellum and parietal lobe, etc. However, in this experiment, we used a 64-channel EEG device with a small number of channels. Not enough to trace back to the cerebellar regions and the inner regions of the brain.
Comment 9: The frequency band related to fatigue is mainly theta frequency band, while alpha frequency band is selected for network attribute analysis in this study, why other frequency bands are not analyzed?
Response 9: Thank you for your comments. Yes, the frequency bands related to fatigue are mainly theta frequency bands. In fact, according to our research content, we selected theta, alpha and beta three major frequency bands for network attribute analysis, and we uploaded relevant experimental data. Please see in Supplementary file 3.
Comment 10: How does lack of sleep affect exercise? I feel that what the author said in the conclusion is not clear and thorough!
Response 10: Thank you for your comments. The effects of lack of sleep on exercise are multi-dimensional. According to your suggestions, we discussed the mechanism of negative effects of lack of sleep on isometric contraction in the discussion, and rewrote the conclusion to clarify our point of view. Please check. Thank you. Please see the red words in the discussion section of the revised manuscript.

Reviewer 2 Report
Comments and Suggestions for Authors
The enclosed manuscript “Changes in muscle-brain connectivity during isometric contraction post-sleep deprivation: based on phase locked value and partial directed coherence” is a research article dealing with the interesting topic of how neuromuscular coupling is affected when no sufficient rest is reached. The thematic of the article is noteworthy as understanding how lack of sleep is affecting muscle functions could have different implications in sport science and medicine. However, the text should be revised due to some misleading statements and language should be extensively edited. The introduction lacks a strong theoretical foundation explaining the rationale behind the study and the main hypothesis. The discussion primarily restates the methods and results, incorporating background information that would be more appropriate in the introduction. Additionally, a neurophysiological interpretation of the findings is missing, which would enhance the impact of the research. Finally, the claim regarding the correlation of handedness with the results is misleading, as the study does not include proper controls for left-handed participants. Therefore, conclusions about handedness should be reconsidered.
Specific comments are listed below:
Abstract:
- The abstract is well-structured but could be more concise. The mention of specific software used for data processing is unnecessary and could be omitted to allow space for expanding the background and discussion.
- Lines 25–26 are overly vague and should be more specific.
- The term "brain muscles’ electrical conduction” (line 30) is unclear. A more precise definition is needed.
Introduction:
- A proper functional definition of brain fatigue should be given.
- Surface myolectricity is a term that is not widely used. Would suggest using electromyography instead.
- “It is well known that surface myoelectricity (sEMG) is a study of the recruitment of motor units by the cerebral cortex during muscle movement, and that α-motor neurons, which act as the α-motor neurons of the randomly moving skeletal muscles, are innervated by the cerebral cortex, and that proprioceptive sensations of the muscles during contraction will be fed back to the sensory centres of the cerebral cortex, thus sEMG is intrinsically related to EEG”. This sentence is not clear and imprecise, the neuromotor control is far more complex. What do the authors mean by randomly moving skeletal muscles? Moreover, saying that the alpha motor neurons are innervated by the cerebral cortex is wrong.
- There is no background information on why the study focuses specifically on athletes
- Aim and main hypothesis are missing.
Methods:
- How did the authors cheked for health and good quality sleep of the participants? Whats a normal body fat percentage and which whre the certain conditions that were used as exclusion criteria? Why only male subjects and why only right handed? All these should be explained and detailed. Moreover the ethical comeettee protocol number is missing.
- Which guidelines were used for the sEMG sensor placement and why the high pass filter was set at 150Hz, knowing that the frequency content of the EMG signal could span until 450 Hz?
- Line 209. What’s motor sensation? Line 239. What is muscle perception? Terminology should be revised and improved throughout all the text.
- No information on artefact rejection is provided when processing EEG signal.
- The choice of network metrics should be justified in terms of their relevance to neurophysiology
- Formulas of the used metrics should be provided.
- More details on network-based-statistics should be provided.
Results:
- Other interesting results on the MF might have arousen if higher frequency were included.
- Was sleep quality assed after aweakening the participants of the two groups? Even in non-sleep deprived subjects sleep might have been disrupted by aby uncontrolled reason.
- Figure 3. The y-axis title should be rephrased.
- I woould avoid repetition of the methodological steps in the result section. Only results should be reported.
- Lines 298-300 are very generic. Please go deeper on assessing these results.
- Figure 4. Why are there lines that are thicker than others? The resting state network should be displayed as well. What are the dashed arrows representing?
- Choice of the frequency bands of interest is not justified. Why did the authors focus only on the gamma, leavign the beta out?
- Figure 5 caption is confusing as it is not explaining deeply what we are observing. Whay are tehre some balls in blu and other in red? What do the arrows directions tell ass?
Discussion and conclusion:
- Discussion as it is, is mainly a resumee of the methods and results. There are many parts that should be moved in the introduction (lines 404-412, 416-426), and others in the methods (lines 446-449). Moreover, there is no need to repropose the data analysis pipeline in this section. Discussion should be compare the obtained findings with the current literature and provide speculation on the physiology underling the affection of neuromechanical coupling after sleep-deprivation.
- In line 478 the authors claim that there is a strong correlation of the results with handeness. How did the authors measured for this correlation? Moreover, a control group is necessary for confirming the statement.
The text should be revised due to some misleading statements and language should be extensively edited.
Reviewer 3 Report
Comments and Suggestions for Authors
The study is devoted to the relationship between the quality of the biceps muscle response to stress during sleep deprivation. The work is written very clearly with the use of adequate references to the literature. At the same time, there are some comments, both significant and those that I would like to leave to the discretion of the authors, which would help improve the quality of this wonderful work.
Critical:
-The method of selecting volunteers (male, right-handed) implies the obligatory reflection of this fact both in the title and in the abstract.
- Fig. 1 "And" - which means, perhaps a typo "The end", then it would be more appropriate to use "Start-Finish" terminology with the designation of intermediate points, if it is something else, then please give a description.
-line 256, the authors talk about interpolation of missing values, it is necessary to indicate how many and what values ​​of parameters could not be obtained, what was the interpolation mechanism; in the discussion section, explain the reasons. It is necessary to answer the question - are the parameter samples normally distributed without supplementing them with interpolated values. If a positive decision is made to publish, the authors could share the data, which would indicate which data are true and which are interpolated. I adequately treat the moderate and justified method of using interpolation of experimental data.
- conduct a critical analysis of approaches and results with similar studies (https://doi.org/10.3389/fnetp.2023.1168677, https://pmc.ncbi.nlm.nih.gov/articles/PMC6143346/, https://doi.org/10.1038/s41598-023-40138-0, https://doi.org/10.1159/000518691)
For consideration (authors decision)
- In the introduction, the authors describe in detail the effects of sleep deprivation on human physiology, however, in my opinion, reinforcing the applicability of the study to real-life situations will help the reader understand the work on a deeper level, and will give the authors possible new directions for further development of the study. I recommend that authors collect literature statistics on the impact of deprivation and related physiological reactions on the occurrence of dangerous life situations: road accidents (10.1186/s12916-018-1025-7 etc), extreme sports (https://pmc.ncbi.nlm.nih.gov/articles/PMC9792101/ etc)
-"impact" line 439 does not seem very usable, muscle-brain communication seems more acceptable.
English Language acceptable in a present quality
Round 2
Reviewer 1 Report
Comments and Suggestions for Authors
I think the authors have answered all my questions very well
Author Response
Comment 1: I think the authors have answered all my questions very well.
Response 1: Thank you for your comments on behalf of all authors.
Reviewer 2 Report
Comments and Suggestions for Authors
The authors submitted an improved version of the manuscript and addressed all the comments provided by the reviewer. Some details must be clarified before consideration for publication.
Plesase, clearly state that the same group has been tested in sleep-deprivation condition and with good sleep condition. In the manuscript current state, this is not cleat to the readers. Moreover, the time period between each experimental session should be provided.
Definition of EMG signal is still imprecise. Surface electromyography does not measure the bioelectrical activity of the spinal motor neurons, rather the action potential of the contracting muscle fibers. When quoting the SENIAM guidelines, please cite the related SENIAM project (http://www.seniam.org/) and related articles.
Author Response
The authors submitted an improved version of the manuscript and addressed all the comments provided by the reviewer. Some details must be clarified before consideration for publication.
Comment 1: Plesase, clearly state that the same group has been tested in sleep-deprivation condition and with good sleep condition. In the manuscript current state, this is not cleat to the readers. Moreover, the time period between each experimental session should be provided.
Response 1: Thank you for your comments and suggestions. We have clearly stated in both the abstract and methodology sections that "the same group has been tested in both sleep-deprived and well-rested states" and “the time period” between models and “the time intervals” between isometric contractions of the left and right biceps brachii test.
Please refer to the green font on Line 18-20, Line 124-134 and Line 136-139.
Comment 2: Definition of EMG signal is still imprecise. Surface electromyography does not measure the bioelectrical activity of the spinal motor neurons, rather the action potential of the contracting muscle fibers. When quoting the SENIAM guidelines, please cite the related SENIAM project (http://www.seniam.org/) and related articles.
Response 2: Thank you very much for your valuable comments and guidance, we have corrected the definition of EMG. Please see the green font on Line 51-52 in revised manuscript. Furthermore, we have cited the authors' original publication in accordance with the SENIAM project (http://www.seniam.org/). Kindly review the green-highlighted text on Line 143-147 and reference [26] in revised manuscript.
[26] Hermens, H.; Frerisk, B. SENIAM. The state of the Art on Sensors and Sensor Placement Procedures for Surface ElectroMy-oGraphy: A proposal for senor placement procedures. Roessingh Research and Development 1997. https://www.researchgate.net/publication/265407503.
